# Improving Dialogue Discourse Parsing via Reply-to Structures of Addressee Recognition

**Yaxin Fan**[1], **Feng Jiang**[2,3,4], **Peifeng Li**[1]*, **Fang Kong**[1], and **Qiaoming Zhu**[1]

[1]School of Computer Science and Technology, Soochow University, Suzhou, China
[2]School of Data Science, The Chinese University of Hong Kong, Shenzhen, China
[3]Shenzhen Research Institute of Big Data, Shenzhen, China
[4] University of Science and Technology of China, Hefei, China
yxfansuda@stu.suda.edu.cn, jeffreyjiang@cuhk.edu.cn
{pfli, kongfang, qmzhu}@suda.edu.cn

## Abstract

Dialogue discourse parsing aims to reflect the relation-based structure of dialogue by establishing discourse links according to discourse relations. To alleviate data sparsity, previous studies have adopted multitasking approaches to jointly learn dialogue discourse parsing with related tasks (e.g., reading comprehension) that require additional human annotation, thus limiting their generality. In this paper, we propose a multitasking framework that integrates dialogue discourse parsing with its neighboring task addressee recognition. Addressee recognition reveals the reply-to structure that partially overlaps with the relation-based structure, which can be exploited to facilitate relation-based structure learning. To this end, we first proposed a reinforcement learning agent to identify training examples from addressee recognition that are most helpful for dialog discourse parsing. Then, a task-aware structure transformer is designed to capture the shared and private dialogue structure of different tasks, thereby further promoting dialogue discourse parsing. Experimental results on both the Molweni and STAC datasets show that our proposed method can outperform the SOTA baselines. The code will be available at https://github.com/yxfanSuda/RLTST.

## 1 Introduction

Dialogue discourse parsing aims to construct a dependency tree on discourse relations to reflect the implicit discourse structure of a dialogue, which is helpful for various downstream tasks, such as reading comprehension (He et al., 2021), meeting summarization (Feng et al., 2021), response generation (Hu et al., 2019) and sentiment analysis (Sun et al., 2021).

Most studies on discourse parsing usually focus on monologue and represent a document with a hierarchical tree on the Rhetorical Structure Theory

---
*Corresponding author

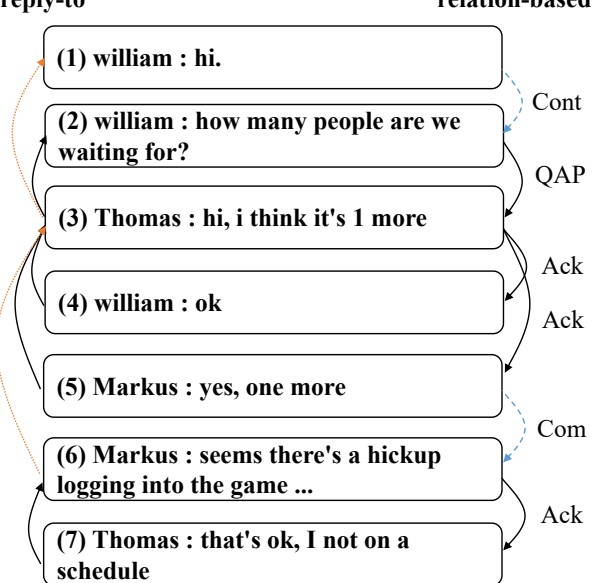

Figure 1: An example of relation-based dialogue discourse parsing from STAC (right lines) (Asher et al., 2016), where *Cont*, *QAP*, *ACK*, and *Com* refer to *Continuation*, *Question-answer_Pair*, *Acknowledgement* and *Comment*, respectively. We also present the reply-to structure of addressee recognition (left lines) in this example for better illustration. Dialogue discourse parsing reveals the **relation-based** structure by establishing discourse links on discourse relations, while addressee recognition reveals the **reply-to** structure by identifying the addressee for each utterance. The black-solid lines indicate the shared structure between the two tasks and the orange-dot lines and the blue-dashed lines indicate the private structure of addressee recognition and dialogue discourse parsing, respectively.

(RST) (Mann and Thompson, 1988). Otherwise, dialogue discourse parsing represents the dialogue as a dependency tree on the Segmented Discourse Relation Theory (SDRT) (Asher and Lascarides, 2003) and an example is shown in Figure 1.

Dialogue discourse parsing suffers from data sparsity because annotating the links and relations requires a precise understanding of dialogues, which is time-consuming and expensive. To mitigate the problem of data sparsity, previous works

explore the complementary knowledge of related tasks to facilitate dialogue discourse parsing, including joint training with reading comprehension (He et al., 2021) or dropped pronoun recovery (Yang et al., 2021). However, the requirements of additional human annotation, such as the questions and answers for reading comprehension, limit the generality.

In this paper, we jointly learn dialogue discourse parsing with its neighbor task addressee recognition (Ouchi and Tsuboi, 2016; Zhang et al., 2018; Le et al., 2019) to alleviate the issue of data sparsity and additional human annotation requirements. Addressee recognition reflects the reply-to structure between different speakers, while dialogue discourse parsing reveals the relation-based structure between the same or different speakers. Therefore, the two tasks have both shared and private structures as shown in Figure 1. Our intuition is to leverage shared structures of addressee recognition to promote dialogue discourse parsing in a multitasking manner.

However, two challenges remain when jointly training our task with addressee recognition. The first challenge arises from the fact that examples from addressee recognition are not equally beneficial for discourse parsing due to the partially overlapping structure between the two tasks. It becomes crucial to identify the most useful examples from the data-rich addressee recognition to improve multitask learning. The second challenge involves how to capture shared and private structures of different tasks. As shown in Figure 1, two tasks have both their shared structure (black-solid lines) and their private structure (orange-dot and blue-dashed lines). Thus, capturing the shared and private structure poses an additional challenge in effectively promoting dialogue discourse parsing.

To tackle the first challenge, we introduce a reinforcement-learning agent to identify the training examples from addressee recognition that contribute the most to dialogue discourse parsing. To address the second challenge, we propose a task-aware structure transformer that effectively exploits the shared and private dialogue structures across different tasks. Our approach involves evaluating the importance of each neuron in the structure transformer and categorizing them as either task-shared or task-private based on their significance for each task. Subsequently, the task-shared neurons are updated by all tasks, while the task-private neurons are updated by each specific task individually.

Experimental results on both Molweni and STAC show that our proposed method outperforms the SOTA baselines. Besides, the reinforcement-learning agent can reduce about 80% of the training data of addressee recognition and help dialogue discourse parsing achieve higher performance. In addition, the task-aware structure transformer can effectively capture shared and private structures of different tasks, thus further promoting dialogue discourse parsing.

## 2  Background

### 2.1  Dialogue Discourse Parsing

Dialogue discourse parsing always focuses on multi-party dialogue and can be divided into two types, i.e., single-task and multitask. Those single-task approaches mainly employ various encoding and decoding methods for discourse parsing. Shi and Huang (2019) jointly and alternatively predicted the link and relation by incorporating the historical structure. Wang et al. (2021) adopted a structure transformer to incorporate both the node and edge information of utterance pairs. Liu and Chen (2021) trained a joint model by merging the parsing datasets. Fan et al. (2022) combined different decoding methods for discourse parsing. Yu et al. (2022) proposed a second-stage pre-trained task to enhance speaker interaction.

Multitask approaches focus on exploring the advantages of relevant tasks to facilitate discourse parsing. Yang et al. (2021) jointly trained dialogue discourse parsing with dropped pronoun recovery, and He et al. (2021) jointly trained discourse parsing with reading comprehension. However, both of them need additional annotated information of the relevant tasks on the basis of discourse parsing, which is time-consuming and costly. In this paper, we propose a multitask framework of dialogue discourse parsing and addressee recognition, which does not need any additional annotated labels.

### 2.2  Addressee Recognition

Addressee recognition aims to determine a possible addressee for some utterance. For example, Ouchi and Tsuboi (2016) first proposed the task and created a large corpus for studying. Le et al. (2019) introduced a who-to-whom model to explore the interaction of speaker and utterance. Gu et al. (2021) proposed several self-supervised tasks to learn who says what to whom.

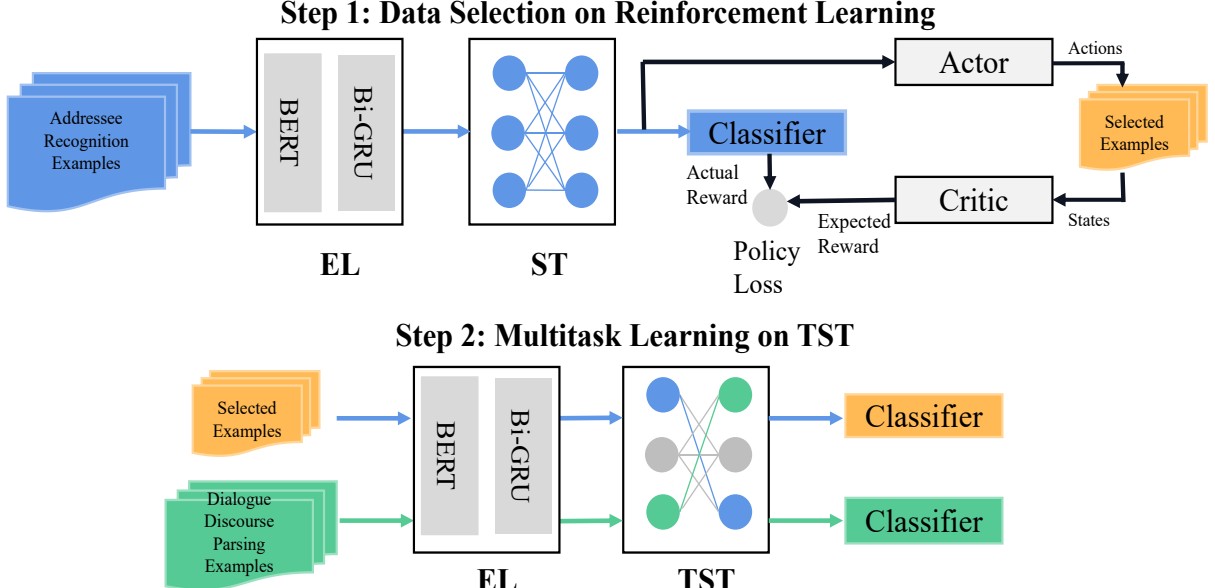

Figure 2: The framework of our method.

## 3 Approach

As shown in Figure 2, our method mainly includes two steps: data selection on reinforcement learning and multitask learning on Task-aware Structure Transformer (TST). In the following sections, we first introduce the backbone of our method, followed by the reinforcement-learning agent and task-aware structure transformer.

### 3.1 Backbone

The backbone includes three components: Encoding Layer (EL), Structure Transformer (ST), and classifier.

#### 3.1.1 Encoding Layer

Given a multi-party dialogue with $n$ utterances denoted as $\{u_1, u_2, \cdots, u_n\}$, we add a dummy root $u_0$ to represent the beginning of a dialogue. The utterances are simply fed into the pre-trained model, such as BERT (Devlin et al., 2019), in the form of $[CLS][SEP]u_0[SEP]u_1\cdots[SEP]u_n[SEP]$ and the hidden states of $[SEP]$ previous to each utterance is regarded as the corresponding utterance representation. Then, a BiGRU was employed on the utterance representation to obtain the dialogue-level context representation $\boldsymbol{H} = \{\boldsymbol{h}_0, \boldsymbol{h}_1, \cdots, \boldsymbol{h}_n\}$, where $\boldsymbol{H} \in \mathbb{R}^{(n+1)\times d}$.

#### 3.1.2 Structure Transformer

Structure Transformer (ST) (Zhu et al., 2019; Liu et al., 2020; Wang et al., 2021) have been widely applied in NLP, which is based on Transformer (Vaswani et al., 2017) and considers both node and edge information at the same time. We feed the dialogue-level context representation $\boldsymbol{H} \in \boldsymbol{H}^F$ into Structure Transformer to obtain the structure representation $\boldsymbol{I}$ as follows.

$$\boldsymbol{z}_{ij}^{(l)} = \frac{(\boldsymbol{W}_q^{(l)}\boldsymbol{h}_i^{(l-1)})(\boldsymbol{W}_k^{(l)}\boldsymbol{h}_j^{(l-1)}) + \boldsymbol{W}_r^{(l)}\boldsymbol{e}_{ij}}{\sqrt{d}} \quad (1)$$

$$\boldsymbol{\alpha}_{ij}^{(l)} = \frac{exp(\boldsymbol{z}_{ij}^{(l)})}{\sum_{t=1}^{n} exp\left(\boldsymbol{z}_{it}^{(l)}\right)} \quad (2)$$

$$\boldsymbol{h}_i^{(l)} = \sum_{j=1}^{N} \boldsymbol{\alpha}_{ij}^{(l)}\left(\boldsymbol{W}_v^{(l)}\boldsymbol{h}_j^{(l-1)} + \boldsymbol{W}_f^{(l)}\boldsymbol{e}_{ij}\right) \quad (3)$$

where $\boldsymbol{h}_i$ and $\boldsymbol{h}_j$ ($\boldsymbol{h}_i, \boldsymbol{h}_j \in \boldsymbol{H}$) are the semantic representations of the utterance $u_i$ and $u_j$ within a dialogue, $\boldsymbol{e}_{ij}$ is the edge representation between $u_i$ and $u_j$. We use three concatenated feature embeddings: speaker, turn, and relative distance as edge information following previous work (Wang et al., 2021), $\boldsymbol{W}_\theta$, where $\theta \in \{q, k, r, v, f\}$, is the learnable parameter. $l$ is the $l$-th layer of Structure Transformer. The dialogue structure representation $\boldsymbol{I} = \{\boldsymbol{h}_0^{(l)}, \boldsymbol{h}_1^{(l)}, \cdots, \boldsymbol{h}_n^{(l)}\}$, $\boldsymbol{I} \in \mathbb{R}^{(n+1)\times d}$.

#### 3.1.3 Classifier

After obtaining the structure representation $\boldsymbol{I}$, we adopt Multi-Layer Perceptrons (MLPs) to calculate probabilities of links between utterance pairs for both tasks and probabilities of relation types for dialogue discourse parsing as follows.

$$S_l, S_r = Softmax(MLP([\boldsymbol{I} : \boldsymbol{I}^T])) \quad (4)$$

where [:] denotes the concatenation operation, $S_l \in \mathbb{R}^{n \times n \times 1}$, $S_r \in \mathbb{R}^{n \times n \times m}$. $n$ and $m$ are the number of utterances within a dialogue and the number of the relation type, respectively.

## 3.2 Data Selection on Reinforcement Learning

The reason for performing data selection is that the partially overlapping structure between both tasks leads to the fact that not all examples in addressee recognition are equally useful for dialogue discourse parsing. Reinforcement learning can automatically select examples from addressee recognition that help discourse parsing the most.

We adopt the actor-critic reinforcement paradigms (Konda and Tsitsiklis, 1999; Ye et al., 2020; Pujari et al., 2022) and add the reinforcement-learning agent to the tail of the backbone. For each example in the addressee recognition training set, we choose the structure representation of dumpy node $h_0^{(l)}$ as the corresponding dialogue representation and feed it into the agent. For a mini-batch $b$ consisting of $m$ dialogues, the semantic representation of $b$ is denoted as $H_b^{(l)}$, where $H_b^{(l)} \in \mathbb{R}^{m \times d}$. We first feed $H_b^{(l)}$ into the actor to decide whether to select the dialogue in mini-batch $b$ and the equation is as follows.

$$P = Softmax(W_a H_b^{(l)} + b_1) \qquad (5)$$

where $P$ is probabilities of actions for dialogues in mini-batch $b$, $W_a$ and $b_1$ are learnable parameters and bias, respectively.

The critic is used to compute the expected reward based on the actions of the actor for the mini-batch $b$ as follows.

$$R = Mean(\sigma(W_b \phi(W_c H_b^{(l)} + b_2) + b_3)) \quad (6)$$

where $R$ is the expected reward of the mini-batch $b$, and $\sigma$ and $\phi$ are the Sigmoid and Tanh activation functions, respectively. $W_b$ and $W_c$ are learnable parameters, respectively, and $b_2$ and $b_3$ are learnable biases.

Then, we assign rewards to the agent by evaluating the performance of discourse links on the validation set of dialogue discourse parsing. If the $F_1$ score on the development set for $g$ mini-batches with size $m$ are denoted as $\{F_1^1, F_1^2, \cdots, F_1^g\}$ and the expected rewards predicted by the critic is denoted as $\{R_1, R_2, \cdots, R_g\}$, the policy loss is computed as follows.

$$\hat{F}_1 = \frac{F_1^i - \mu}{\sigma + \epsilon} \qquad (7)$$

$$L_r = -\frac{1}{g}\sum_{i=1}^{g}(\hat{F}_1 - R_i) \times \frac{1}{m}\sum_{j=1}^{m} log(P[a_j^i])$$
$$+ \frac{1}{g}\sum_{i=1}^{g} L_1(R_i, \hat{F}_1)$$

$$(8)$$

where $\mu$ and $\sigma$ are the mean and standard deviations of the $F_1$ score, $\epsilon$ is a smoothing constant, $a_j^i$ is an action for the $j$-th dialogue of the $i$-th mini-batch decided by the actor, and $L_1$ is the smooth L1 loss. The algorithm for reinforcement learning is shown in Appendix A.

## 3.3 Task-aware Structure Transformer

Under multitask learning, the structure transformer can only capture the shared structure of different tasks, because either the examples from addressee recognition or dialogue discourse parsing equally update all neurons of $W_\theta$ in the structure transformer at the back-propagation step.

To capture shared and private structures of different tasks, we proposed Task-aware Structure Transformer (TST) based on a structure transformer. Inspired by the task of model pruning (Zhu and Gupta, 2018; Evci et al., 2020), we assume that different neurons in $W_\theta$ have different importance to different tasks. Based on this hypothesis, we can divide the neurons into two types: task-shared and task-private. We force those task-shared neurons to participate in the shared structure learning of all tasks while those task-private neurons only focus on some specific tasks. Thus, the key is to determine the task-shared and task-private neurons in $W_\theta$ for each task. To this end, we use a binary mask matrix $M \in \{0, 1\}^{|W_\theta|}$ to indicate task-shared and task-private neurons for each task. Each element in $M$ corresponds to a neuron in $W_\theta$. An element of 0 indicates that the neuron is task-shared while 1 indicates that the neuron is task-private.

To obtain a mask matrix $M$, we first train the structure transformer on all tasks in a multitasking manner to obtain the initial learnable parameters $W_\theta^0$. Then, we fine-tune $W_\theta^0$ on a specific task to evaluate the importance of neurons, and the most important $\alpha\%$ of neurons are reserved as the task-private neurons of the task. Intuitively, fine-tuning $W_\theta^0$ on a specific dataset will amplify the magnitude of the gradient of the important neurons. Thus, we rank the gradient of neurons in descending order and reserve the most important $\alpha\%$ of neurons as the task-private neurons. $M$ is obtained by set-

ting the indices of task-private neurons to 1 and the others to 0. Once the mask matrix $M^F$ for all tasks is obtained, we continue to fine-tune the structure transformer on all tasks and only update task-private neurons for each task during the back-propagation step. The whole process is shown in Algorithm 1.

---

**Algorithm 1** TST Learning

---

**Require:** addressee recognition dataset $A$, dialogue discourse parsing dataset $D$, initial learnable parameters $W_\theta^0$ of structure transformer, and learning rate $\eta$.

1: **for** dataset $c$ in $\{A, D\}$ **do**
2:     initialize mask $M^c = \{0\}^{|W_\theta^0|}$;
3:     fine-tuning $W_\theta^0$ on dataset $c$ and obtain gradients $G^c$ for all neurons in $W_\theta^0$
4:     sort gradients $G^c$ in a descending order and obtain the indices $I^c$ of top $\alpha\%$ gradients
5:     set $M_i^c = 1$ if $i \in I^c$
6: **end for**
7: initialize time step $t = 1$
8: **for** epoch $e = 1, 2, \cdots$ **do**
9:     **for** dataset $c$ in $\{A, D\}$ **do**
10:         **for** mini-batch $j$ in dataset $c$ **do**
11:             $W_\theta^t = W_\theta^{t-1} - \eta \cdot G^c \odot M^c$
12:             $t = t + 1$
13:         **end for**
14:     **end for**
15: **end for**
16: **return** $W_\theta^t$

---

### 3.4 Multitask Learning

For optimization of addressee recognition, we minimize the cross-entropy of golden links as follows.

$$L_a(\theta_a) = -\sum_{i=1}^{n} y^* log P(y_i) \qquad (9)$$

where $\theta_a$ denotes the parameters of addressee recognition to be optimized, $y^*$ represents the golden links, $y_i$ is the predicted links, and $n$ is the number of golden links.

For optimization of dialogue discourse parsing, we minimize the cross-entropy of golden links and relation types as follows.

$$L_b(\theta_b) = -\sum_{i=1}^{n} y^* log P(y_i) - \\ \sum_{i=1}^{n}\sum_{j=1}^{m} r_{ij}^* log P(r_{ij}) \qquad (10)$$

| Task | Dataset | Train | Valid | Test | UttNum |
|------|---------|-------|-------|------|--------|
| DDP | STAC | 1062 | - | 111 | 1-37 |
|     | Molweni | 9000 | 500 | 500 | 7-14 |
| AR | Hu | 311K | 5K | 5K | 6,7 |
|    | Ou5 | 461K | 28K | 32K | 5 |
|    | Ou10 | 495K | 30K | 35K | 10 |
|    | Ou15 | 489K | 30K | 35K | 15 |

Table 1: Statistics of Dialogue Discourse Parsing (DDP) and Addressee Recognition (AR). 'UttNum' indicates utterance number.

where $\theta_b$ denotes the parameters of dialogue discourse parsing to be optimized, $y^*$ and $r_{ij}^*$ represent golden links and relation types, respectively. $y_i$ and $r_{ij}$ represent predicted links and relation types, respectively. $n$ is the number of golden links and $m$ is the number of relation types.

For multitask learning, we add the loss of all tasks as follows.

$$L = L_a + L_b \qquad (11)$$

## 4 Experimentation

### 4.1 Datasets

We evaluate our proposed model on two dialogue discourse datasets STAC (Asher et al., 2016) and Molweni (Li et al., 2020) and the statistics are shown in Table 1. For STAC, we follow previous work and select 10% of the training dialogues for validation. For addressee recognition, we utilize all training data from four commonly used corpora Hu (Hu et al., 2019) and Ou5/Ou10/Ou15 (Ouchi and Tsuboi, 2016), including about 1.75 $M$ dialogues, to facilitate dialogue discourse parsing in a multitasking manner.

### 4.2 Experimental Settings

We use *bert-based-uncased* to initialize the parameters of BERT. The learning rate of BERT, reinforcement learning, and structure transformer are set to 1e-5, 3e-5, and 3e-4, respectively. The layer and head of the structure-aware transformer are set to 1 and 4, respectively. Performance of adopting different values of alpha in TST is shown in Appendix B and we set $\alpha$ to 0.8 and 0.7 on Molweni and STAC, respectively. The maximum utterance number for Molweni and addressee recognition dataset are set to 15. We follow previous work (Wang et al., 2021) and adopt a sliding window to split the dialogue in STAC into chunks and the maximum utterance

| Model | | Molweni | | STAC | |
|---|---|---|---|---|---|
| | | **Link** | **Link&Rel** | **Link** | **Link&Rel** |
| Single-task | ChatGPT | 59.91 | 25.25 | 63.75 | 23.85 |
| | DSM | 76.94 | 53.49 | 71.99 | 53.62 |
| | SSAM | 81.63 | 58.54 | 73.48 | 57.31 |
| | COMB | 80.15 | 56.60 | 73.25 | 57.18 |
| | DAMT | 82.50 | 58.91 | 73.64 | 57.42 |
| | SSP | 83.70 | 59.40 | 73.00 | 57.40 |
| | Single-ST | 81.06 | 56.81 | 71.24 | 55.51 |
| Multitask | DiscProReco | - | - | **74.10** | 57.00 |
| | DPRC | 80.00 | 57.00 | - | - |
| | Multi-ST$_{Full}$ | 82.59 | 58.21 | 72.31 | 56.49 |
| | Multi-ST$_{Subset}$ | 83.16 | 58.80 | 73.06 | 57.24 |
| Ours | TST$_{Full}$ | 83.75 | 59.21 | 73.26 | 57.47 |
| | TST$_{Subset}$ | **85.26** | **60.91** | 73.69 | **57.63** |

Table 2: Main results on both Molweni and STAC.

number for STAC is set to 40. The maximum sequence length for Molweni, STAC, and addressee recognition dataset are set to 380, 512, and 380. The batch sizes for Molweni, STAC, and addressee recognition dataset are set to 200, 100, and 200 respectively. The epoch for reinforcement learning, structure transformer, and task-aware structure transformer are set to 5, 3, and 3, respectively. All experiments are performed using a GeForce RTX 3090 GPU. In this paper, we follow Shi and Huang (2019) and adopt *Link* $F_1$ and *Link&Rel* $F_1$ for evaluation. The *Link* $F_1$ evaluates the performance of link prediction, and *Link&Rel* $F_1$ evaluates the performance of link and relation prediction.

## 4.3 Baselines and Experimental Results

We compare our method with the following baselines. **Single-task**: 1) **ChatGPT**(Fan and Jiang, 2023): it adopt ChatGPT[1] for dialogue discourse parsing. 2) **DSM** (Shi and Huang, 2019), it alternately predicted discourse links and discourse relations by incorporating historical structure; 3) **SSAM** (Wang et al., 2021), it adopted a structure transformer and two auxiliary training signals for parsing; 4) **COMB** (Liu and Chen, 2021), it trained the model by merging the parsing datasets; 5) **DAMT** (Fan et al., 2022), it combined transition-based and graph-based decoding methods for parsing; 6) **SSP** (Yu et al., 2022), it proposed a second-stage pre-trained task to enhance the speaker interaction; 7) **Single-ST** (our simplified version), it utilized BERT and BiGRU to encode the dialogues and applied a structure transformer to cap-

ture the structure of the single dialogue discourse task. **Multitask**: 1) **DiscProReco** (Yang et al., 2021), it proposed to jointly train dropped pronoun recovery and dialogue discourse parsing; 2) **DPRC** (He et al., 2021), it proposed to jointly train reading comprehension and dialogue discourse parsing; 3) **Multi-ST**$_{Full}$ (our simplified version), it used all addressee recognition datasets and employed a structure transformer to capture the shared structure between addressee recognition and dialogue discourse parsing. 4) **Multi-ST**$_{Subset}$ (our simplified version), Similar to Multi-ST$_{Full}$ but using the selected data from all addressee recognition datasets on reinforcement learning.

As shown in Table 2, we report the main results on the Molweni and STAC test sets following previous work (Fan et al., 2022; Yu et al., 2022). our TST$_{Subset}$ achieves a *Link* $F_1$ of 85.26 and a *Link&Rel* $F_1$ of 60.91 in the Molweni test set, outperforming all previous SOTA systems. On STAC, our TST$_{Subset}$ achieves 73.69 on *Link* $F_1$ and 57.93 on *Link&Rel* $F_1$, outperforming most of the previous SOTA systems. These results show the effectiveness of our TST$_{Subset}$ on dialogue discourse parsing and ensure that dialogue discourse parsing can benefit from addressee recognition.

Compared with Single-ST, our simplified version of Multi-ST is effective for joint training dialogue discourse parsing with addressee recognition. In addition, Multi-ST$_{Subset}$ can achieve better performance than Multi-ST$_{Full}$ by using about 20% and 19% training data of addressee recognition on Molweni and STAC, respectively, indicating the effectiveness of data selection on reinforcement

---

[1]https://openai.com/blog/chatgpt

| Addressee Recognition | Dialogue Discourse Parsing | | | |
| --- | --- | --- | --- | --- |
| | Molweni | | STAC | |
| | Link | Link&Rel | Link | Link&Rel |
| None | 81.06 | 56.81 | 71.24 | 55.51 |
| Hu | 82.45 | 57.87 | 72.02 | 56.35 |
| Ou5 | 81.48 | 56.95 | 71.34 | 55.57 |
| Ou10 | 81.96 | 57.20 | 71.69 | 55.62 |
| Ou15 | 82.16 | 57.32 | 71.76 | 55.80 |
| Subset | **83.16** | **58.80** | **73.06** | **57.24** |

Table 3: Performance on dialogue discourse parsing in ablation study for each addressee recognition dataset. 'None' indicates without data of addressee recognition and 'Subset' indicates the selected data from all addressee recognition datasets on reinforcement learning.

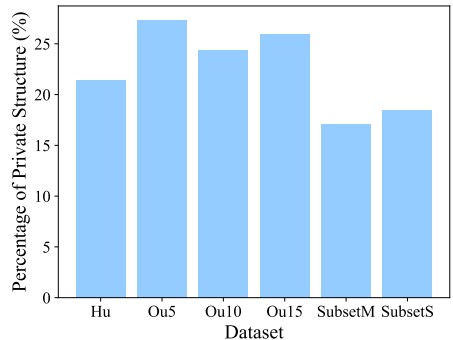

Figure 3: Percentage of private structures in different addressee recognition datasets. The private structures are as shown orange-dot lines in Figure 1. 'SubsetM' and 'SubsetS' indicates the subsets selected from addressee recognition on reinforcement learning for dialogue discourse parsing dataset Molweni and STAC, respectively.

learning. However, Multi-ST can only capture the shared structure and neglect the influence of private structures of addressee recognition on dialogue discourse parsing. Compared with Multi-ST, our TST$_{Subset}$ captures shared and private structures of different tasks, which can significantly improve the performance of dialogue discourse parsing. Notably, our approach does not need any additional annotated information, compared with DiscProReco or DPRC. These results verify the effectiveness of our method.

Our TST$_{Subset}$ performs worse on STAC than on Molweni, with the same trend as other SOTA systems. We attribute this to the fact that more utterances of dialogue in STAC increase the difficulty of dialogue discourse parsing as shown in Table 1. In addition, the performance of our TST$_{Subset}$ has a slight decrease on *Link* $F_1$ on STAC, in comparison with DiscProReco. We attribute this to the different pre-trained models and joint learning tasks. Moreover, TST$_{Full}$ only achieves comparable performance with the previous systems on the task of addressee recognition because of the small size of the dialogue discourse parsing dataset.

## 5 Analysis

### 5.1 Effectiveness of Data Selection

**Ablation Study of addressee recognition datasets** To reveal the effectiveness of adopting reinforcement learning for data selection, we conduct ablation studies on the four addressee recognition datasets, as shown in Table 3. We can see that all addressee recognition datasets can improve the performance of discourse parsing and the Hu dataset contributes the most. Subsets on reinforcement learning that only select around 20%

and 19% data from all four datasets for Molweni and STAC, respectively, help dialogue discourse parsing achieve higher performance. On Molweni, the performance reaches 83.16 and 58.80 on *Link* $F_1$ and *Link&Rel* $F_1$ metrics, respectively. On STAC, the performance reaches 73.06 and 57.24 on *Link* $F_1$ and *Link&Rel* $F_1$, respectively. These results indicate the effectiveness of data selection by using reinforcement learning.

**Less private structures of addressee recognition help more** Since *reply-to* structures of addressee recognition and *relation-based* structures of discourse parsing partially overlap, we hypothesize that examples with less private structures in addressee recognition can better benefit dialogue discourse parsing. To validate this, we analyze the percentage of private structures in different datasets, as shown in Figure 3. Among all addressee recognition datasets, Hu has the smallest proportion of private structures, accounting for 21%. This corresponds to the results in Table 3, where Hu benefits discourse parsing the most. In addition, the subsets selected for Molweni and STAC have an even lower proportion of private structures, accounting for 17% and 19%, respectively. This is consistent with our intuition that the addressee recognition dataset with less private structure can facilitate dialogue discourse parsing better.

### 5.2 Effectiveness of TST

**Ablation study of TST** To illustrate the effectiveness of TST, we compare it with several variants as shown in Table 4. *Prune-ST* indicates that we reserve the task-private neurons for each task and

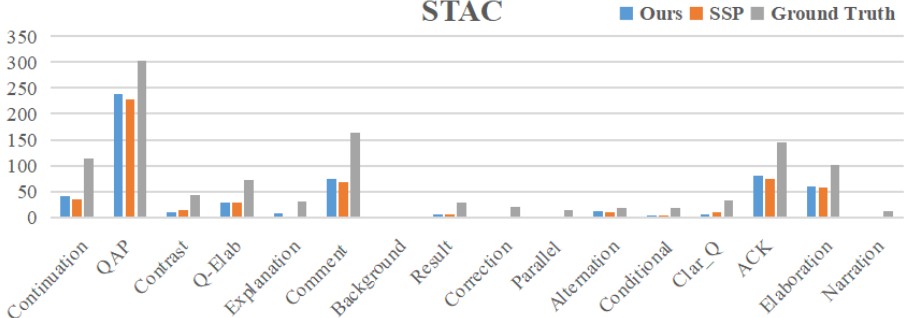

Figure 4: The comparison of correct relation prediction between our TST$_{Subset}$ and SOTA system SSP on STAC.

| Model | Molweni | | STAC | |
|---|---|---|---|---|
| | Link | Link&Rel | Link | Link&Rel |
| Prune-ST | 65.15 | 33.32 | 61.04 | 32.87 |
| Random-ST | 82.29 | 57.85 | 72.23 | 56.50 |
| Reversed-ST | 79.15 | 53.87 | 70.82 | 52.09 |
| TST | **85.26** | **60.91** | **73.69** | **57.63** |

Table 4: Ablation study of Link and Link&Rel F$_1$ performance of TST. Subsets selected from addressee recognition was adopted for all experiments.

| | Accuracy | | | |
|---|---|---|---|---|
| Method | Molweni | | STAC | |
| | Shared | Private | Shaerd | Private |
| None | 80.13 | 54.49 | 63.01 | 71.61 |
| Multi-ST$_{Full}$ | 81.60 | 53.90 | 64.38 | 70.33 |
| Multi-ST$_{Subset}$ | 82.00 | 57.48 | 65.34 | 72.12 |
| TST$_{Full}$ | 83.33 | 59.87 | 64.66 | 73.17 |
| TST$_{Subset}$ | **84.46** | **63.47** | **65.48** | **75.45** |

Table 5: Accuracy of shared and private structures in dialogue discourse parsing. The shared and private structures are shown in black-solid and blue-dashed lines in Figure 1, respectively.

abandon the shared neurons. *Random-ST* indicates that we randomly select $\alpha$ % of specific neurons for each dataset. *Reversed-ST* indicates that we rank the gradients of neurons and reserve the lowest $\alpha$ % neurons as task-private neurons. We can see that the performance decreases sharply when adopting *Prune-ST* in comparison with our TST, which indicates the essential of shared neurons. When adopting *Random-ST* or *Reversed-ST*, the performance decreases significantly in comparison with our TST, which indicates the effectiveness of our approach that reserving the top $\alpha$ % neurons according to the gradients as task-private neurons.

**TST can significantly capture both shared and private structures** To explore whether TST can

capture both shared and private structures significantly, we analyze the accuracy of both shared and private structures in dialogue discourse parsing, as shown in Table 5. We can see that Multi-ST$_{Full}$ can promote the shared structure performance but lead to a slightly decline in private structure. We attribute this to the negative effect of private structures of addressee recognition on the private structure learning of dialogue discourse parsing. Besides, Multi-ST$_{Subset}$ can promote the performance of private structures in discourse parsing by reducing private structures in addressee recognition. Furthermore, our TST demonstrates the capability to promote the performance of shared and private structures in dialogue discourse parsing through joint learning with either the *Full* data of addressee recognition or a *Subset* selected from addressee recognition using reinforcement learning. These results prove the effectiveness of our TST.

### 5.3 Analysis on Improvement of Relation Performance

Given that relation is correct if and only if both link and relation type is predicted correctly, the improvement in relation performance can be attributed to the success of link prediction. To explore what kinds of relations benefit the most from our method, we analyze the number of correct relation predictions of STAC as shown in Figure 4. We compare our method TST$_{Subset}$ with the SOTA system SSP (Yu et al., 2022) that uses the same pretrained model as us. We can see that our method outperforms the baseline on some high-resource relations like *Question-Answer_Pair* (QAP), *Comment*, *Acknowledgement* (ACK), *Continuation* and *Elaboration*. The improvement of these relations mainly derives from the success of link prediction. Besides, both our TST$_{Subset}$ and SSP hardly predict the low-resource relations, indicating that the low-resource relations cannot benefit from the im-

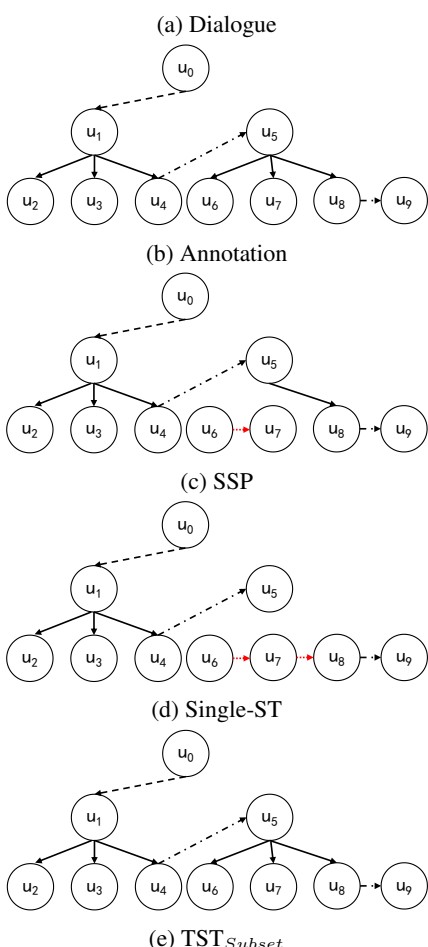

($u_0$) A: anyone have any idea why ubuntu would boot correctly after hanging during one boot , and visa versa ?
($u_1$) B:when it hangs , what does it say before hanging itself ?
($u_2$) A: in fact , i think it will boot after hibernating
($u_3$) A: it hangs on a blank ( `` dead '' ) screen
($u_4$) A: it appears to be just as gnome is loading
($u_5$) B: can you get a login prompt by pressing ctrlaltf1 when it hangs ?
($u_6$) A: no , and ctrl-alt-del will not reboot , either
($u_7$) A: nothing happens it still says network:0 disabled when i push the button
($u_8$) A: when i reboot the computer manually to recover from the hang , it boots fine
($u_9$) B: can you copy FILEPATH to a pastie ?

(a) Dialogue

(b) Annotation

(c) SSP

(d) Single-ST

(e) TST$_{Subset}$

Figure 5: (a) is a dialogue from Molweni. (b) is the discourse structure annotated manually. (c)-(e) are structures predicted by SSP, Single-ST, and our TST$_{Subset}$, respectively. Solid, dashed, dashed-dotted, and red dotted lines denote the relations *Question-Answer_Pair*, *Question_Elaboration*, *Clarification_Question*, and *Continuation*, respectively.

provement of link prediction, and more attention is required in the future. The same phenomenon can be observed on Molweni, as shown in Appendix C.

## 5.4 Case Study

To better illustrate the advantage of our method, we draw the discourse structures predicted by several baselines and our method. The dialogue and discourse structures are shown in Figure 5. In the annotation, we can observe that $u_6$, $u_7$, and $u_8$ that all uttered from speaker A form the *Question_Answer Pair* relation with $u_5$ uttered from speaker B, which is shared with the reply-to structure of addressee recognition. Although SSP enhanced the speaker interaction, $u_6$-$u_7$ is still recognized incorrectly. Besides, our simplified version of Single-ST only considers the relation-based structure of dialogue discourse parsing, leading to the incorrect recognition $u_6$-$u_7$ and $u_7$-$u_8$. However, our TST$_{Subset}$ recognizes $u_5$-$u_6$, $u_5$-$u_7$, and $u_5$-$u_8$ correctly by leveraging the reply-to structure of addressee recognition in a multitasking manner, which illustrates the effectiveness of our method.

## 6 Conclusion

In this paper, we proposed to improve dialogue discourse parsing by leveraging reply-to structures of addressee recognition in a multitasking manner. To this end, we first adopt a reinforcement-learning agent to identify training examples from addressee recognition that help dialogue discourse parsing the most. Then, we proposed a task-aware structure transformer to capture the shared and private structures of different tasks. Experimental results on Molweni and STAC verify the effectiveness of our method. In the future, we will focus on how to incorporate multiple relevant tasks to improve dialogue discourse parsing.

## Limitations

Our work has several limitations, which we aim to address in our future work. First, since our method is to promote the target task with the data of the neighbor task, we can certainly use the data of the dialogue discourse parsing to promote addressee recognition. However, due to the much smaller size of the corpus of dialogue discourse parsing than addressee recognition, the performance of the addressee recognition is not improved significantly. Second, our method focuses on the improvement of link prediction, and the improvement of relation prediction is mainly derived from the success of link prediction due to the cascade nature. In the future, we will continue to refine our approach to overcome the above shortcomings.

## Acknowledgements

The authors would like to thank the three anonymous reviewers for their comments on this paper. This research was supported by the National Natural Science Foundation of China (Nos. 62376181, 62276177, and 62276178), and Project Funded by the Priority Academic Program Development of Jiangsu Higher Education Institutions (PAPD).

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

## A  Algorithm of Data Selection on Reinforcement Learning

Algorithm 2 illustrates the process of selecting examples from addressee recognition by reinforcement learning. For each example of addressee

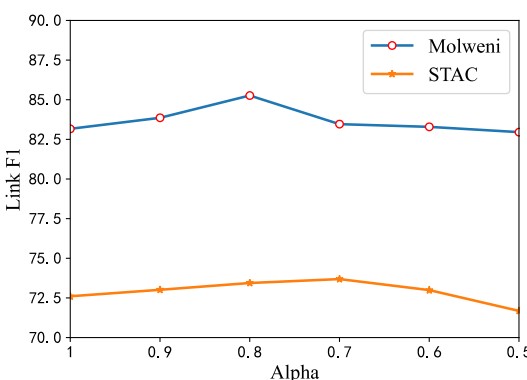

Figure 6: Link F1 on the test set of Molweni and STAC with different values of alpha.

recognition, the actor-network $AN$ determines whether or not to select and the critic-network computes the expected reward $R^{dev}$ based on actions of the actor. Then, the selected examples are used to train the backbone $BN$ and obtain actual reward $F_1^{dev}$. Finally, the parameters of actor-network $AN$ and critic-network $CN$ are updated. The selected examples $\{b_0^{select}, b_1^{select}, \cdots\}$ are returned after training.

---

**Algorithm 2** Data Selection on Reinforcement Learning

---

**Require:** addressee recognition dataset $A$, validation set of dialogue discourse parsing $D^{dev}$, backbone $BN$, actor network $AN$, and critic network $CN$.

1: **for** epoch $e = 1, 2, \cdots$ **do**
2:     **for** mini-batch $b$ in $A$ **do**
3:         $AN$ make select or not decision for each example in $b$
4:         $CN$ computes expected reward $R^{dev}$ based on examples selected by $AN$
5:         train $BN$ on the selected mini-batch subset $b^{select}$
6:         evaluate $BN$ on $D^{dev}$ and obtain $F_1^{dev}$
7:         compute reinforcement loss based on $F_1^{dev}$ and $R^{dev}$
8:         update the parameters of actor network $AN$ and critic network $CN$
9:     **end for**
10: **end for**
11: **return** $\{b_0^{select}, b_1^{select}, \cdots\}$

---

## B  Performance of TST with Different Values of Alpha

The performance on the test set of Molweni and STAC as alpha changes as shown in Figure 6. We

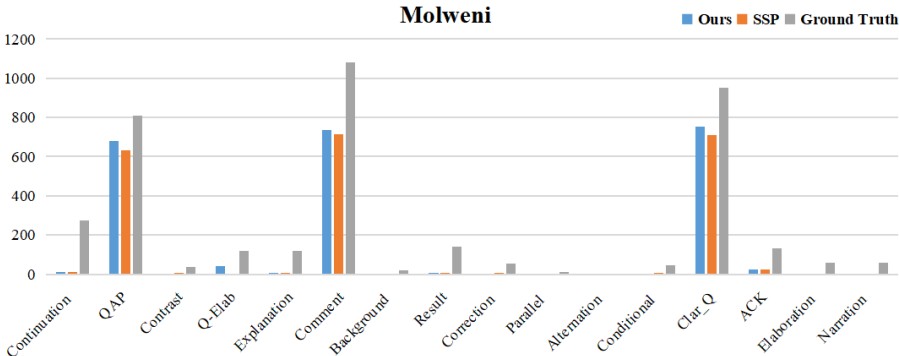

Figure 7: The comparison of correct relation prediction between our TST$_{Subset}$ and SOTA system SSP on Molweni.

can see that the performance of link prediction reaches the peak when alpha is set to 0.8 and 0.7 on Molweni and STAC, respectively.

## C    Relation Performance on Molweni

Figure 7 shows the performance of relation on Molweni. We can observe that our method outperforms the baseline on some high-resource relations like *Question-Answer_Pair* (QAP), *Comment*, and *Clarification_question* (Cla_q). However, both our TST$_{Subset}$ and SSP hardly predict the low-resource relations.