# OpenReview forum: "Improving Dialogue Discourse Parsing via Reply-to Structures of Addressee Recognition"
_EMNLP/2023/Conference — EMNLP 2023 Main_

### Official Review · Reviewer_hgfu · 2023-07-28

**Soundness:** 4

**Excitement:**

4: Strong: This paper deepens the understanding of some phenomenon or lowers the barriers to an existing research direction.

**Paper Topic And Main Contributions:**

This paper proposes a multitasking framework that integrates dialogue discourse parsing with addressee recognition to improve the performance of dialogue discourse parsing. The paper addresses the problem of data sparsity in dialogue discourse parsing, which requires precise understanding of dialogues and is time-consuming and expensive.

The main contributions of the paper are:
1. Proposing a reinforcement learning agent to identify training examples from addressee recognition that are most helpful for dialogue discourse parsing.
2. Designing a task-aware structure transformer to capture the shared and private dialogue structure of different tasks, thereby further promoting dialogue discourse parsing.
3. Outperforming the state-of-the-art baselines on both the Molweni and STAC datasets.

**Reasons To Accept:**

The strengths of this paper are:

1. The proposed multitasking framework integrates dialogue discourse parsing with addressee recognition, which does not require any additional annotated labels and can alleviate the problem of data sparsity in dialogue discourse parsing.
2. The paper proposes a reinforcement learning agent and a task-aware structure transformer to improve the performance of dialogue discourse parsing.
3. The proposed method outperforms the state-of-the-art baselines on both the Molweni and STAC datasets.

If this paper were to be presented at the conference or accepted into Findings, the main benefits to the NLP community would be:

1. The proposed multitasking framework can be applied to other related tasks to facilitate discourse parsing and improve the performance of NLP models.
2. The reinforcement learning agent and task-aware structure transformer can be used in other NLP tasks to capture shared and private structures of different tasks and improve the performance of NLP models.
3. The proposed method can be used to develop more accurate and efficient dialogue systems, which can benefit various applications such as customer service, personal assistants, and chatbots.

**Reasons To Reject:**

N/A

**Reproducibility:**

4: Could mostly reproduce the results, but there may be some variation because of sample variance or minor variations in their interpretation of the protocol or method.

**Reviewer Confidence:**

4: Quite sure. I tried to check the important points carefully. It's unlikely, though conceivable, that I missed something that should affect my ratings.

---

> ### Author Rebuttal · Authors · 2023-08-26
>
> Thank you very much for your positive comments!

---

### Official Review · Reviewer_AseC · 2023-08-03

**Typos Grammar Style And Presentation Improvements:** Missing the description of *ACK* in t…
**Soundness:** 4

**Excitement:**

4: Strong: This paper deepens the understanding of some phenomenon or lowers the barriers to an existing research direction.

**Missing References:**

N/A

**Paper Topic And Main Contributions:**

This paper attempts to improve the performance of discourse parsing by jointly optimizing the neighboring task, addressee recognition, in a multitasking training manner. The motivation is that the *reply-to* structure in the addressee recognition task is similar to the relation-based structure in the discourse parsing task. To achieve the goal, a two-step method is presented. The first step is to select the most suitable examples of addressee recognition via a reinforcement learning agent. The second step is to jointly optimize the two tasks via a task-aware structure Transformer. Experiments on STAC and Molweni datasets show the effectiveness of the proposed method. Comprehensive ablation studies and analyses are also provided.

**Questions For The Authors:**

Question A. You use the gradient as the metric to divide the task-private and shared neurons. Why do you choose *gradient* and what other factors can be considered?

Question B. The joint training of discourse parsing and addressee recognition can improve the performance of discourse parsing. So, does the joint training also benefit to the performance of addressee recognition?

Question C. How do you think the reasons that ChatGPT performs extremely worse?



**Reasons To Accept:**

1. The paper is clearly written and well organized. It is easy for readers to understand and follow up.

2. Straightforward and effective method. It is a common solution to adopt multi-task learning to jointly improve the dialogue discourse parsing and the related task. Advanced than the common solution, the authors propose the important example selection and the task-aware structure Transformer is well-motivated and effective.

3. Comprehensive evaluations.

**Reasons To Reject:**

I do not see any rejection reasons so far.  But I have questions that waiting for the author response.

**Reproducibility:**

4: Could mostly reproduce the results, but there may be some variation because of sample variance or minor variations in their interpretation of the protocol or method.

**Reviewer Confidence:**

3: Pretty sure, but there's a chance I missed something. Although I have a good feel for this area in general, I did not carefully check the paper's details, e.g., the math, experimental design, or novelty.

---

> ### Author Rebuttal · Authors · 2023-08-26
>
> Thank you very much for your valuable comments and suggestions.
>
> **Q1: Why do you choose gradient as metrics and what other factors can be considered?**
>
> A1: There are various factors used to assess neuron significance, such as weight, gradient, activation, and activation sparsity. Neuron weight and gradient are commonly used and effective methods in previous studies. We conducted preliminary experiments using both weight and gradient methods and found that the gradient performs better. The potential explanation is that gradient information can reveal the neurons’ adaptability of the model to the dialogue data, thus enabling a more accurate representation of the data's characteristics.
>
> **Q2: Does the joint training benefit the performance of addressee recognition?**
>
> A2: Our model only achieves comparable performance with the previous systems on addressee recognition. We attribute this to the small size of the dialogue discourse parsing dataset as illustrated in lines 420-423. We suggest that joint training can enhance the performance of addressee recognition  if the scale of the dialogue discourse parsing dataset approaches that of the addressee recognition dataset.
>
> **Q3: The reasons for ChatGPT performs extremely worse.**
>
> A3: As the cited paper [1] pointed out, dialogue discourse parsing is a relatively complex task which can be treated as a spanning tree problem and it is difficult for ChatGPT to understand the rhetorical structures. Moreover, some work [2, 3] also pointed out that ChatGPT performs poorly in some complex tasks. In addition, we have two private views. Firstly, the seq2seq paradigm (the next token is generated based on previous tokens, and the closer tokens have a more significant impact.) of ChatGPT is linear, both for pretraining and instruction tuning. Thus, ChatGPT can understand the linear topic structures better, but performs poorly in hierarchical discourse structures, as shown in the experimental results in the cited paper [1]. Secondly, it could be that no similar task to discourse parsing in the phase of instruction tuning, leading to the poor performance in the out of domain scenario.
>
> **Q4: Missing the description of ACK in the caption of Figure 1.**
>
> A4: Thanks for your careful review. ACK is short for Acknowledgement. We will introduce it in our revised paper.
>
> [1] Fan, Y. and Jiang, F., 2023. Uncovering the potential of ChatGPT for discourse analysis in dialogue: an empirical study. arXiv preprint arXiv:2305.08391.
>
> [2] Liu, H., Ning, R., Teng, Z., Liu, J., Zhou, Q. and Zhang, Y., 2023. Evaluating the logical reasoning ability of ChatGPT and gpt-4. arXiv preprint arXiv:2304.03439.
>
> [3] Gao, J., Zhao, H., Yu, C. and Xu, R., 2023. Exploring the feasibility of ChatGPT for event extraction. arXiv preprint arXiv:2303.03836.

---

### Official Review · Reviewer_nH36 · 2023-08-11

**Typos Grammar Style And Presentation Improvements:** n/a
**Soundness:** 3

**Excitement:**

3: Ambivalent: It has merits (e.g., it reports state-of-the-art results, the idea is nice), but there are key weaknesses (e.g., it describes incremental work), and it can significantly benefit from another round of revision. However, I won't object to accepting it if my co-reviewers champion it.

**Missing References:**

n/a

**Paper Topic And Main Contributions:**

To tackle the data sparsity issue for dialogue discourse parsing, this paper utilizes datasets from addressee recognition in a multi-task setting to boost the performance of dialogue discourse parsing.

**Questions For The Authors:**

- Could you provide some insights/numbers on how dialogue discourse parsing performance will impact downstream tasks?
- Could you provide more details on multi-task training for Equation 11, as these two tasks have datasets at different scales?

**Reasons To Accept:**

- The proposed framework utilizes large addressee recognition datasets (>300K conversations) to mitigate the data sparsity issue for the dialogue discourse parsing (<10K conversations) task, and improves the state-of-the-art performance on this task.
- The approach might inspire people working on other tasks regarding utterance relations.

**Reasons To Reject:**

- The contribution is incremental. The paper mentioned the purpose of dialogue discourse parsing is to benefit downstream tasks like reading comprehension, response generation, and sentiment analysis. However, it's unclear how significant the marginal improvement is on these downstream tasks, which do not suffer from data sparsity issues.

**Reproducibility:**

4: Could mostly reproduce the results, but there may be some variation because of sample variance or minor variations in their interpretation of the protocol or method.

**Reviewer Confidence:**

3: Pretty sure, but there's a chance I missed something. Although I have a good feel for this area in general, I did not carefully check the paper's details, e.g., the math, experimental design, or novelty.

---

> ### Author Rebuttal · Authors · 2023-08-26
>
> Thank you very much for your valuable comments and suggestions.
>
> **Q1: Could you provide some insights/numbers on how dialogue discourse parsing performance will impact downstream tasks?**
>
> A1: The specificity of dialogue lies in the segmentation of sentences (utterances) between parties in communication. These utterances are not independent but rather are part of a larger dialogue discourse unit. Thus, the structure of discourse between utterances, particularly in multi-party dialogues, can reveal the significance of historical discourse structure in a dialogue and is a crucial foundation for downstream tasks, thereby improving their performance.
>
> Previous research has demonstrated that modeling dialogue discourse structures can enhance downstream tasks. Regarding response generation, Hu et al. [1] argue that it is inadequate to encode utterances sequentially, as many real-world dialogues involve multiple interlocutors. They propose a Graph-Structured neural Network to capture directly the discourse structures in multi-party dialogues. Experimental results reveal that incorporating discourse structures can result in a 2.27 BLEU enhancement on Ubuntu Dialogue Corpus. Regarding meeting summarization, Feng et al. [2] propose that it is preferable to model the varying relations among distinct utterances rather than considering the meeting as a linear progression. They present a Dialogue Discourse-Aware Meeting Summarizer explicitly modeling the interaction between utterances in a meeting by modeling different discourse relations parsed by the Deep Sequential Model (DSM) [3]. By integrating discourse structures, there was a 1.67 and 0.64 improvement in ROUGE-L on AMI and ICSI, respectively. In sentiment analysis, Sun et al. [4] argue that being aware of dialogue discourse structures is better than only capturing their sequential information. They propose a discourse-aware graph neural network to model these structures. Ablation studies show that discourse structures can improve Average-F1 by 1.88, 1.78, and 1.00 on MELD, EmoryNLP, IEMOCAP, respectively.
>
> **Q2: More details on multi-task training for Equation 11.**
>
> A2: In this paper, we set the loss weight ratio for both tasks as 1:1, consistent with the approach used by He et al. [5]. Although we also experimented with different weight ratios, our preliminary results suggested no significant improvement in the parsing task. Therefore, we adopt the multi-task training of He et al. [5] and refrain from further exploring weighted loss optimization.
>
> [1] Hu, W., Chan, Z., Liu, B., Zhao, D., Ma, J. and Yan, R., 2019. GSN: A graph-structured network for multi-party dialogues. arXiv preprint arXiv:1905.13637.
>
> [2] Feng, X., Feng, X., Qin, B. and Geng, X., 2020. Dialogue discourse-aware graph model and data augmentation for meeting summarization. arXiv preprint arXiv:2012.03502.
>
> [3] Shi, Z. and Huang, M., 2019, July. A deep sequential model for discourse parsing on multi-party dialogues. In Proceedings of the AAAI Conference on Artificial Intelligence (Vol. 33, No. 01, pp. 7007-7014).
>
> [4] Sun, Y., Yu, N. and Fu, G., 2021, November. A discourse-aware graph neural network for emotion recognition in multi-party conversation. In Findings of the Association for Computational Linguistics: EMNLP 2021 (pp. 2949-2958).
>
> [5] He, Y., Zhang, Z. and Zhao, H., 2021. Multi-tasking dialogue comprehension with discourse parsing. arXiv preprint arXiv:2110.03269.

---

### Meta-Review · Area_Chair_f2eA · 2023-09-12

**Recommendation:** 4

**Metareview:**

The paper presents a multitasking framework that combines dialogue discourse parsing and addressee recognition to address the issue of data sparsity in dialogue discourse parsing. It introduces a reinforcement learning agent to identify the most beneficial training examples from addressee recognition, and a task-aware structure transformer to optimize the performance for both tasks. The method has shown superior performance over state-of-the-art baselines on the Molweni and STAC datasets.

Pros:
The multitasking approach doesn't need extra annotated labels so it alleviates data sparsity in dialogue discourse parsing.
The incorporation of a reinforcement learning agent enhances dialogue discourse parsing.
State-of-the-art performance on two datasets.
The techniques proposed can be potentially applied to other NLP tasks, benefiting various applications like chatbots and personal assistants.
The paper is described as being well-written, organized, and straightforward.

Cons:
The significance of the improvements for downstream tasks, especially those without data sparsity challenges, is not well-demonstrated.

---

### Decision · Program_Chairs · 2023-10-07

**Decision:**

Accept-Main

**Comment:**

The paper presents a multitasking framework that combines dialogue discourse parsing and addressee recognition to address the issue of data sparsity in dialogue discourse parsing. It introduces a reinforcement learning agent to identify the most beneficial training examples from addressee recognition, and a task-aware structure transformer to optimize the performance for both tasks. The method has shown superior performance over state-of-the-art baselines on the Molweni and STAC datasets.

Pros:
The multitasking approach doesn't need extra annotated labels so it alleviates data sparsity in dialogue discourse parsing.
The incorporation of a reinforcement learning agent enhances dialogue discourse parsing.
State-of-the-art performance on two datasets.
The techniques proposed can be potentially applied to other NLP tasks, benefiting various applications like chatbots and personal assistants.
The paper is described as being well-written, organized, and straightforward.

Cons:
The significance of the improvements for downstream tasks, especially those without data sparsity challenges, is not well-demonstrated.